# Obstruction of the Hepatic Venous Flow Caused by Intravenous Leiomyomatosis

**DOI:** 10.3390/medicina56120696

**Published:** 2020-12-14

**Authors:** Sin-Youl Park, In Hwan Yeo, Yun Jeong Kim, Jong Kun Kim

**Affiliations:** 1Department of Emergency medicine, College of Medicine, Yeungnam University, Daegu 42415, Korea; dryuri@naver.com; 2Department of Emergency Medicine, Kyungpook National University School of Medicine, Daegu 41404, Korea; inani@hanmail.com (I.H.Y.); kimyjem1@gmail.com (Y.J.K.)

**Keywords:** Budd–Chiari syndrome, Inferior vena cava, leiomyoma, leiomyomatosis, ultrasonography

## Abstract

Budd–Chiari syndrome (BCS) is a rare intrahepatic vascular disease that is characterized by a hepatic venous outflow obstruction. Intravenous leiomyomatosis (ILs) is a rare complication of a myoma. Here, we report a case of BCS that was caused by intracaval ILs. A woman presented to the emergency department (ED) with abdominal distension that had gradually progressed over a period of 3 years. Bedside ultrasonography and contrast-enhanced computed tomography (CECT) showed a large ascites and pelvic mass. The mass continued to the inferior vena cava and the right atrium. The intracaval mass was obstructing the left and middle hepatic veins. We established a tentative diagnosis of BCS caused by intracaval ILs and attempted surgical resection. Complete resection of the intracaval mass failed because of adhesion; however, she was discharged from the hospital without any postoperative complications. After 3 months, a pelvic ultrasonography showed a recurrence of a 4 × 3 cm pelvic mass. The mass size increased to 6 cm after 30 months. ILs can cause secondary BCS and can lead to life-threatening conditions. Owing to its extreme rarity, early detection in the ED is challenging. Bedside ultrasonography and CECT can enable the early recognition of BCS by ILs.

## 1. Introduction

Myoma is the most common type of gynecological tumor among pre-menopausal middle-aged women. Intravenous leiomyomatosis (ILs) is a rare complication of a myoma, characterized by venous invasion of myoma tissue [1]. Since its introduction in 1896, it has been rarely reported as an individual case report or series [2]. Its pathology is unclear, and the vascular wall origin theory and uterine leiomyoma invading theory exist [1]. Although histologically benign, ILs can extend into the inferior vena cava, the right side of the heart, and even the pulmonary artery. The clinical manifestations of ILs depend on the site and extent of the involvement, and in severe cases can lead to death if untreated [3].

Budd–Chiari syndrome (BCS) is a rare heterogeneous group of intrahepatic vascular diseases characterized by a hepatic venous outflow obstruction. It manifests as the following three characteristic symptoms: abdominal pain, hepatomegaly, and ascites, and can be life-threatening if appropriate treatment is not administered [4,5].

Since the first case was reported in 1996 by Kuenen et al., only two cases of BCSs as complications of ILs have been reported in the English-language literature [6,7,8]. We report a case of BCS detection as a complication of ILs, through ultrasonography and contrast-enhanced computed tomography (CECT) in a patient who presented to the emergency department (ED) with abdominal distension.

## 2. Case Presentation

A female patient in her early 50 s with abdominal pain and distension visited the ED. She had first experienced abdominal pain and distension 3 years previously; however, the patient was not treated, owing to financial limitations. The abdominal distension had gradually worsened in the 3 months before her presentation to the ED. She had been diagnosed with uterine myoma 15 years previously; however, she did not undergo any medical treatment. She consumed about 1 bottle of soju for 2 weeks and took no medications other than sleeping pills. The patient’s history constituted two gravida and two abortions, irregular periods, and dysmenorrhea. She started menstruating at 14 years of age. There was usual hypermenorrhea; however, she had not menstruated in the previous 2 months.

On presentation, her blood pressure was 140/80 mmHg, pulse rate was 74 bpm, and oxygen saturation was 96%; she was afebrile. On physical examination, we found that her abdomen was severely distended, and she had pitting edema on both legs. Moreover, right lung auscultation was decreased. The laboratory results included a hemoglobin concentration of 10.9 g/dL and platelet count of 373 K/μL. Serum profiles, including blood chemistry, electrolyte, and liver and renal function tests were normal; however, her albumin level was 2.69. Proteinuria was observed on urinalysis. Chest radiography showed right pleural effusion.

Bedside ultrasonography for abdominal distension showed a large amount of ascites and right pleural effusion, and hepatomegaly with an irregular surface was visible (Figure 1a). A 10 × 9-cm hyperechoic homogenous mass was observed in the pelvic cavity.

Two hours after admission, her blood pressure suddenly dropped to 80/60 mmHg. Bedside ultrasonography was re-implemented, and saline was administered rapidly. There was no interval change in the abdominal findings. However, the inferior vena cava (IVC) was scanned for volume measurement, and a hyperechoic mass was found in the IVC (Figure 1b). The mass continued to the right heart and moved back and forth from the right atrium (RA) to the right ventricle (RV) as per the heartbeat (Appendix A). The mass was attached to the right anterior wall of the RA-IVC function, and as it descended, it became smaller in size and appeared to float freely (Appendix A).

After hydration, the patient’s blood pressure rose to 100/80 mmHg. Consistent with the bedside ultrasonography findings, the CECT showed large amounts of ascites and pleural effusion, multiple lobulated pelvic cavity mass, and mass in the IVC and the heart (Figure 2). The hypodense mass in the IVC was observed from the iliac bifurcation level to the right heart. The liver was shown as inhomogeneous mottle. This intracaval mass was located in the orifice of the left and middle hepatic veins, causing obstruction. In the delay phase, left and middle hepatic venous flow was observed to drain through the collateral to the dilated right hepatic vein.

We established a tentative diagnosis of BCS caused by ILs with IVC and heart extension and decided to perform surgical resection using a team approach. Thoracic, gynecology, and vascular surgeons participated in the surgery. First, a total abdominal hysterectomy with bilateral salpingo-oophorectomy was performed by the gynecological surgeons. A pelvic mass measuring about 11.5 × 7.5 × 6.0 cm was removed and identified as a myoma on biopsy (Figure 3).

Thereafter, intracardiac and intracaval tumor removal was performed by thoracic and vascular surgeons. A well-capsulated intracardiac mass was freely movable in the RA and RV; however, the intracaval tumor could not move freely and densely adhered to the RA-IVC joint, and blind traction was considered dangerous; therefore, debulking surgery was performed. The formal report of histopathology of thoracic and vascular specimens showed the following results. The first specimen consisted of a piece of an intracardiac mass (mass of the right atrium) (69.3 g, 7.0 × 5.0 × 3.8 cm), and visual examination showed a hard, homogeneous tan mass with a swirling pattern. There was no bleeding or necrosis. It was judged to be a thrombus. A second specimen consisting of a piece of pericardial tissue (1.7 × 0.8 cm) could be diagnosed as a benign smooth muscle tumor consistent with intravenous leiomyomatosis. The third specimen consists of three pieces of intravascular mass (10.6 g, 3.3 × 2.6 × 0.8 cm) and can be diagnosed as intravenous leiomyomatosis (Figure 4).

After surgery, aspartate aminotransferase, alanine aminotransferase, and direct bilirubin increased to 1306 IU/L, 1053 IU/L, and 6.21 mg/dL, respectively, but on the 28th day of hospitalization, improved to 75 IU/L, 97 IU/L, and 0.21 mg/dL, respectively. The patient was discharged without complications 1 month postoperatively. On pelvic ultrasonography that was performed after 3 months, a 4 × 3-cm pelvic mass had recurred (Figure 5).

Furthermore, 30 months postoperatively, pelvic ultrasonography showed that the tumor size had increased to 6 cm. Since then, the patient has not visited the outpatient clinic.

## 3. Discussion

In the diagnosis of BCS, clinical suspicion is most significant. However, because its clinical presentations are similar to those of liver and biliary tract diseases, which are common diseases in the ED, establishment of a diagnosis of BCS by symptoms in the ED is challenging.

The clinical presentation of BCS depends on the extent and rapidity of hepatic vein occlusion and venous collateral circulation. Asymptomatic patients account for about 15% of the total patients, and about 10% show clinical symptoms of acute diseases and visit the hospital [9,10]. The abnormalities for liver function are similar to those seen in chronic liver disease or cirrhosis [11]. The serum aspartate and alanine aminotransferase levels may be markedly elevated in the fulminant and acute forms of BCS, while the increases may be smaller in the subacute and chronic forms.

Bedside ultrasonography is a useful imaging tool for the discrimination of causes of abdominal distension, shortness of breath, and shock in the ED. Because bedside ultrasonography is useful to detect pelvic masses and intracaval and/or intracardiac status, and can evaluate hepatic vein flow, it can be an appropriate initial imaging tool for the early recognition of ILs and BCS by ILs [12]. CECT is useful to detect myoma tissue in the pelvis, IVC, and heart. In addition, it is good for overall assessment of the invading site of ILs and it is useful for delineating the venous anatomy and liver configuration. Thus, CECT is useful for preoperative diagnosis or comprehensive evaluation of BCS by ILs.

MRI is the most accurate way to detect, localize, and characterize myoma. In ILs, MRI can be useful for distinguishing between tumors and blood clots [13]. However, MRI generally does not provide more information than CECT in the diagnosis of ILs. MRI is generally not essential for IL diagnosis except in complex or special cases.

Thus far, no standard treatment has been established for BCS; however, expert groups have suggested relief in the hepatic venous outflow duct obstruction as the main treatments [14]. The natural course of untreated patients with Budd-Chiari syndrome is unknown. The 5-year survival rate is 75% after surgery, and with radiation intervention, the 5-year survival rate is 83%; however, with only medical therapy and without surgical or radiological intervention, the 5-year survival rate is around 44% [15].

The key to the treatment of ILs as a cause of BCS is the surgical resection of tumors, although no definite treatment has been established due to its rarity [16]. Surgical resection is performed by multidisciplinary cooperation among gynecology, cardiac surgery, and vascular surgeons by one-stage operation or a staged surgery. Despite the presence of extensive intravascular invasion, they generally have a long-term survival rate after successful removal of the ILs [17]. However, the average recurrence rate is reported to be about 30% [18,19]. Complete resection of ILs and the multidisciplinary cooperation among gynecology, cardiac surgery, and vascular surgeons is essential for the reduction in recurrence.

The recurrence may increase depending on the residual ILs; complete removal of the tumor is the key to preventing recurrence [20]. ILs that grow freely within the blood vessels and/or cardiac chamber can be removed relatively easily. However, complete removal of ILs that are adhesive or invade the wall of the IVC or the heart chamber is difficult. Further, aggressive extraction of ILs can damage the IVC or the heart and cause fatal bleeding [12,21,22].

Hormonal therapy for ILs has been attempted on the rationale that the tumor has hormonal properties similar to uterine myoma [23]. Although ILs exhibit progesterone-dependent properties, the effectiveness of antiprogesterone therapy on ILs is not clear and it is no longer being used as treatment [24].

Anti-estrogen therapy has shown good effects on ILs in some early cases but has not shown consistent evidence of reducing tumor size or improving clinical outcomes for ILs [21,25]. Gonadotropin-releasing hormone (GnRH) agonists can induce a low estrogen state, reducing the size of ILs and preventing recurrence of ILs [26,27,28]. However, ILs may grow again after stopping GnRH agonist treatment [27,29]. Surgical intervention is required if the size of the ILs increases again after discontinuation of GnRH agonist treatment [21,30].

Currently, the primary treatment for ILs is surgical resection, and hormonal therapy cannot replace this treatment. Hormone therapy can be used as an adjuvant therapy to reduce tumor size before surgery or to minimize tumor growth after incomplete resection. It can also be used as an optional treatment for patients who refuse or are unable to have surgery.

## 4. Conclusions

Abdominal distension is a common cause for visits to the ED. ILs, a rare complication of leiomyoma, can cause secondary BCS by obstruction of hepatic out-flow. Although ILs is histologically benign, it can lead to life-threatening conditions. Owing to its extreme rarity, it is difficult to identify BCS by ILs in the ED. Bedside ultrasonography and CECT, initial imaging tools for abdominal distension, may be useful for the early recognition of BCS by IL.

## Figures and Tables

**Figure 1 medicina-56-00696-f001:**
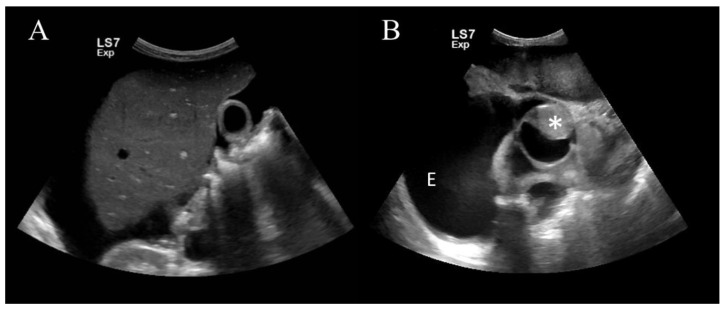
Bedside ultrasonography findings. (**A**) A liver with an irregular surface and a large amount of perihepatic ascites are observed. (**B**) Bedside ultrasonography demonstrates a hyperechoic mass (asterisk) in the inferior vena cava (IVC) and a large amount of pleural effusion (E). The mass is attached to the right anterior wall of the right atrium (RA)-IVC function, and as it descended, it became smaller in size and appeared to float freely.

**Figure 2 medicina-56-00696-f002:**
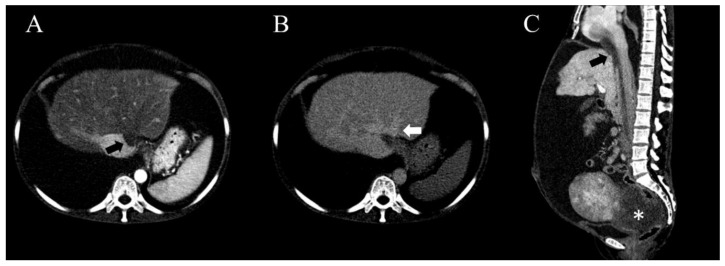
The findings of chest and abdominal contrast enhanced computed tomography. (**A**) In the arterial phase, the liver is shown as inhomogeneous mottle, and hypodense mass (black arrow) is shown in the inferior vena cava (IVC). This mass in the IVC is located in the orifice of the left and middle hepatic veins, causing obstruction of the left and middle hepatic venous flow. (**B**) In the delay phase, the left and middle hepatic venous flow drains through the collateral to the dilated right hepatic vein (white arrow). (**C**) The elongated mass (black arrow) of the IVC is located in the tricuspid valve of RA and RV and is observed almost to the level of the iliac bifurcation. The signal intensity of intracaval mass is similar to that of the pelvic mass (asterisk).

**Figure 3 medicina-56-00696-f003:**
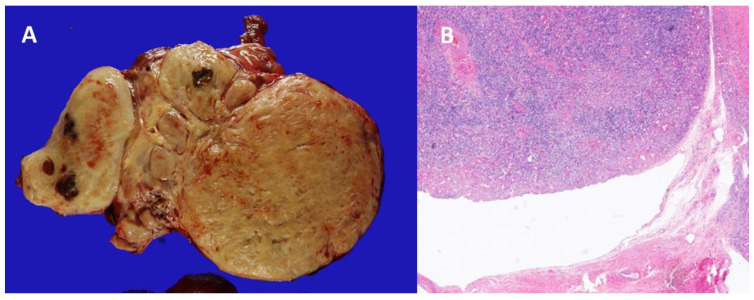
Gross photo and microscopic photo of the hysterectomy specimen. (**A**) An ultilobulating 11.5-cm-sized huge mass was identified at the left posterolateral wall of uterine corpus. The myometrium of uterus was thickened showing trabecular patterned cut surface indicating diffuse adenomyomatosis (black arrowhead). (**B**) Microscopic photo of uterine intravenous leiomyomatosis. The tumor invades into the dilated venous space (original magnification × 20).

**Figure 4 medicina-56-00696-f004:**
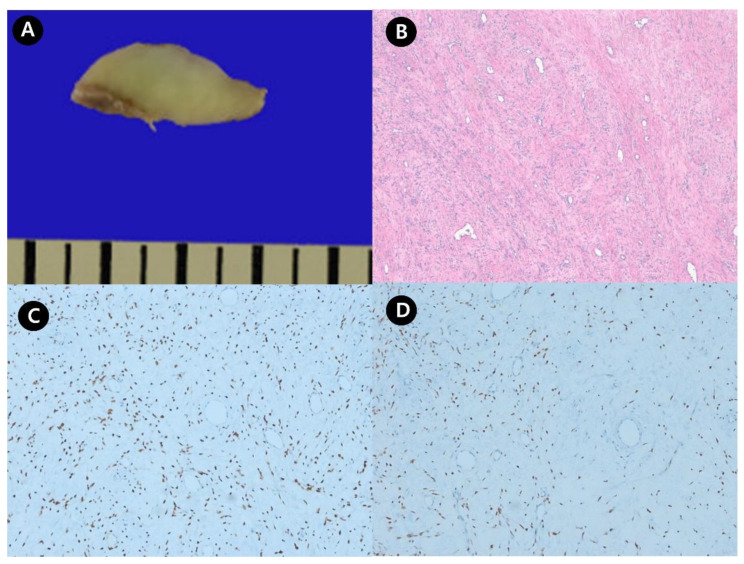
Gross photo and microscopic photo of intravascular mass. (**A**) Gross photo of resected intravenous leiomyoma. (**B**) Microscopic photo of intravenous leiomyoma. Lose spindle cell proliferation with prominent small vascular proliferation is identified (original magnification × 40). (**C**) Immunohistochemical staining for estrogen receptor showed diffuse nuclear positivity of tumor cells (original magnification × 100). (**D**) Immunohistochemical staining for the progesterone receptor showed diffuse nuclear positivity of the tumor cells (original magnification × 100).

**Figure 5 medicina-56-00696-f005:**
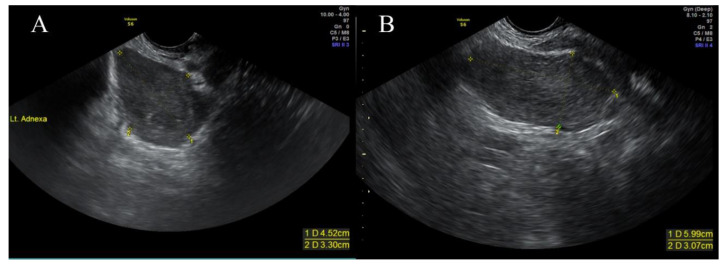
The findings of a follow-up pelvic ultrasonography performed in a gynecology outpatient clinic. (**A**) Pelvic ultrasonography that was performed after 3 months showed that a 4 × 3 cm pelvic mass had recurred. (**B**) The tumor size increased to 6 cm, as shown in pelvic ultrasonography that was conducted 30 months postoperatively.

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
