# Peer review of "Obstruction of the Hepatic Venous Flow Caused by Intravenous Leiomyomatosis"

_medicina, 2020, doi:10.3390/medicina56120696_

Round 1
Reviewer 1 Report
What is the difference between POCUS and US?
What is "the city hall welfare staff"? and is it important to present the case? As you said that they visited the ED, I would like to know what was wrong with them.
discussion:
-you say intravenous invasion of a myoma is rare, but you cite a 10% incidence. Please check the reference. 10% is not rare!
-The role of MRI in case of IL should be discussed.
-anti-progestins are neglected in the discusion
-what about GnRH treatment when surgery is not possible?
-what about survival if surgery is not possible?
Author Response
RESPONSE to Reviewers' comments
We greatly appreciate the academic editor’s comments and suggestions to improve this manuscript.
The authors have commissioned MDPI Author Services for this manuscript's english language and style Editing. We will upload the results as soon as they become available.
Comments and Suggestions for Authors
- What is the difference between POCUS and US?
< Response to comments >
- Thank you for your valuable review.
- The authors defined the USG as a imaging test that systematically scans the requested anatomical site for a sufficient time by transferring the patient to the ultrasound laboratory, and defined the POCUS as after taking a portable usg machine to the patient, intensive and limited scan for a suspected lesion in a short time.
- In manuscript. we revised POCUS to bedside US in considering your point-out.
2.What is "the city hall welfare staff"? and is it important to present the case? As you said that they visited the ED, would like to know what was wrong with them.
< Response to comments >
- Thank you for your valuable review.
- The sentence states that the patient was not feeling well for several months ago but did not visit the hospital due to financial problems, and that the patient visited the hospital with the help of the city hall welfare staff.
- As you pointed out, "the city hall welfare staff" was not an important part, so we modified it in the manuscript.
- A female patient in her early 50s with abdominal pain and abdominal distension visited the ED.
discussion:
- you say intravenous invasion of a myoma is rare, but you cite a 10% incidence. Please check the reference. 10% is not rare!
< Response to comments >
- Thanks for the appropriate point.
- Your point is appropriate, so we corrected as follows, and the meaning was clarified. And added the reference.
- Intravenous leiomyomatosis (ILs) is a rare complication of myoma characterized by venous invasion of myoma tissue(1). Since its introduction in 1896, it has been reported rarely as an individual case report or series (2).
- The role of MRI in case of IL should be discussed.
< Response to comments >
- Thank you for your valuable review.
- Your point is appropriate, so we described MRI role on ILS in discussion as follows.
- MRI is the most accurate way to detect, localize, and characterize myoma. In ILs, MRI can be useful for distinguishing between tumors and blood clots. However, MRI generally does not provide more information than CT in the diagnosis of ILs. MRI is generally not essential for IL diagnosis except in complex or special cases.
- anti-progestins are neglected in the discussion.
< Response to comments >
- Thank you for your attentive review.
- In consideration of your point-out, we described about the anti-progestins for ILs and the reference in the discussion.
- Hormonal therapy for ILs has been attempted on the rationale that the tumor has hormonal properties similar to uterine myoma. Although ILs exhibit progesterone dependent properties, the effectiveness of antiprogesterone therapy on IL is not clear and is no longer being used as treatment. Anti-estrogen therapy has shown good effects on IL in some early cases but has not shown consistent evidence of reducing tumor size or improving clinical outcomes for ILs.
- what about GnRH treatment when surgery is not possible?
< Response to comments >
- Thank you for your valuable review.
- The authors agree with your point and described in the discussion as following.
- Gonadotropin-releasing hormone (GnRH) agonists can induce a low estrogen state, reducing the size of ILs and preventing recurrence of ILs. However, ILs may grow again after stopping GnRH agonist treatment. Surgical intervention is required if the size of the ILs again increases after discontinuation of GnRH agonist treatment. As of now, the primary treatment on ILs is surgical resections, and hormonal therapy cannot replace surgical resection. Hormone therapy can be used as an adjuvant therapy to reduce tumor size before surgery or to minimize tumor growth after incomplete resection. It can also be used as optional treatment for patients who refuse surgery or are unable for surgery.
- what about survival if surgery is not possible?
< Response to comments >
- Thank you for important point-out.
- In consideration of your point-out, we described about survival of BCS and ILs in the discussion. And added related references
- Thus far, no standard treatment has been established for BCS; however, several expert groups have suggested relief in the hepatic venous outflow duct obstruction as the main treatments (12). The natural course of untreated patients with Budd-Chiari syndrome is unknown. The 5-year survival rate is 75% after surgery, and with radiation intervention, the 5-year survival rate is 83%; however, with only medical therapy without surgical or radiological intervention, the 5-year survival rate is around 44%(13).
- The key to the treatment of ILs as a cause of BCS is surgical resection of tumors although no definite treatment has been established for its rarity.
- In respect of ILs as cause of BCS, although it is benign histologically, because ILs can extend into the inferior vena cava, right-side heart, and even the pulmonary artery, in severe cases can lead to death in untreated case.
- As of now, the primary treatment on ILs is surgical resections, and hormonal therapy cannot replace surgical resection. Hormone therapy can be used as an adjuvant therapy to reduce tumor size before surgery or to minimize tumor growth after incomplete resection. It can also be used as optional treatment for patients who refuse surgery or are unable for surgery.
Reviewer 2 Report
Some observations:
- The title seems no correct, is it an uterine myoma that constrict vena cava ? And the intracaval (not intracarval) specimen is simply a thrombus . If it is not so put the histologic slide .
- The multiple operation is very interesting but could be better put some intraoperatory picture of thoracic and vascular part
Author Response
RESPONSE to Reviewers' comments
We greatly appreciate the academic editor’s comments and suggestions to improve this manuscript.
The authors have commissioned MDPI Author Services for this manuscript's English language and style Editing. We will upload the results as soon as they become available.
Comments and Suggestions for Authors Some observations:
- The title seems no correct, is it an uterine myoma that constrict vena cava? And the intracaval (not intracarval) specimen is simply a thrombus. If it is not so put the histologic slide.
< Response to comments >
- Thank you for your valuable review.
- For your point-out, our authors reviewed CT images of patient with OBGY radiologists with more than 10 years of experience. The radiologists confirmed to our authors that CT images showed hepatic venous flow obstruction was caused by the intracaval mass of ILs located in the orifice of the left and middle hepatic veins rather than by constriction of IVC by uterine myoma. In this manuscript, “BCS as complications of ILs” is key part, So, our authors want to keep the title.
- We corrected “intracarval” to “intracaval”
- According to your appropriate point-out, the authors had found the histopathological formal reports on a specimen of this patient and recorded it in the manuscript. For the histologic images, we requested the pathology department the histologic slide, and we attach related images to the manuscript.
- Figure 3. Gross photo of the hysterectomy specimen and microscopic photo of uterine intravenous leiomyomatosis. A. Multilobulating 11.5 cm sized huge mass was identified at the left posterolateral wall of uterine corpus. The myometrium of uterus was thickened showing trabecular patterned cut surface indicating diffuse adenomyomatosis (black arrowhead). B. The tumor invades into dilated venous space (original magnification x 20).
- Figure 4. Gross photo and microscopic photo of intravascular mass. A. Gross photo of resected intravenous leiomyoma. B. microscopic photo of intravenous leiomyoma. Lose spindle cell proliferation with prominent small vascular proliferation is identified (original magnification x 40). C. immunohistochemical staining for estrogen receptor showed diffuse nuclear positivity of tumor cells (original magnification x 100). D. immunohistochemical staining for progesterone receptor showed diffuse nuclear positivity of tumor cells (original magnification x 100).
2. The multiple operation is very interesting but could be better put some intraoperative picture of thoracic and vascular part
< Response to comments >
- Thank you for your valuable review.
- As you pointed out, our authors agree that it would be better if the intraoperative pictures of thoracic and vascular surgery were included to this manuscript, and I looked for pictures. but unfortunately, the intraoperative picture was not archived, so we could not include it in this manuscript.
Round 2
Reviewer 2 Report
I think thath intravascular leiomyoma probably is a primitive caval leiomyoma sincronous with uterine leyomiomatosis.It Is very interesting but the title is no correct
Author Response
RESPONSE to Reviewers' comments
We greatly appreciate the academic editor’s comments and suggestions to improve this manuscript.
Comments and Suggestions for Authors Some observations:
- I think thath intravascular leiomyoma probably is a primitive caval leiomyoma sincronous with uterine leyomiomatosis.It Is very interesting but the title is no correct.
< Response to comments >
- Thank you for your valuable review.
- As your point-out, our authors modified the title to “Obstruction of the hepatic venous flow caused by intravenous leiomyomatosis”.